# Do the Pelvic and Thorax Movements Differ between the Sexes and Influence Golf Club Velocity in Junior Golfers?

**DOI:** 10.3390/sports11030060

**Published:** 2023-03-08

**Authors:** Tomas Gryc, Frantisek Zahalka, Matěj Brožka, Jitka Marenčáková, Petr Miřátský, Arnold Baca, Michael Stöckl

**Affiliations:** 1Sport Science Laboratory, Faculty of Physical Education and Sport, Charles University, 162 52 Prague, Czech Republic; 2Department of Biomechanics, Kinesiology and Computer Science in Sport, Institute of Sport Science, University of Vienna, 1150 Wien, Austria

**Keywords:** golf swing, skilled golfers, driving performance, ball velocity, junior golfers

## Abstract

The aim of this study was to determine the differences in golf swing execution in terms of the parameters of the pelvis and thorax movement between the sexes in junior golfers and their relation to the golf club velocity. Elite female and male players (age: 15.4 ± 1.0 and 15.8 ± 1.7 years, respectively) performed 10 golf swings with a driver under laboratory conditions. Pelvis and thorax movement parameters and golf club velocities were measured using a three-dimensional motion capture system. Statistical parametric mapping analysis of pelvis–thorax coupling revealed a significant difference (*p* < 0.05) between boys and girls during backswing. Analysis of variance showed a significant effect of sex on the parameters of maximal pelvic rotation (F = 6.28, *p* = 0.02), X-factor (F = 5.41, *p* = 0.03), and golf club velocity (F = 31.98, *p* < 0.01). No significant relationship was found between pelvis and thorax movement parameters and golf club velocity in the girls. We found a significant negative relationship between the parameters of maximal thorax rotation and golf club velocity (r = −0.941, *p* < 0.01) and between X-Factor and golf club velocity (r = −0.847, *p* < 0.05) in the boys. We suggest that these negative relationships in males were caused by the influence of hormones during their maturation and biological development, where there is decreased flexibility (lower shoulders rotation and X-factor) and growth of muscle strength (higher club head velocity).

## 1. Introduction

In the Czech Republic, the numbers of registered male and female golfers (men, 64.2%) and teaching professionals (men, 90.9%) are disproportionate. The golf swing technique literature intended for golf professionals [1] and the literature on the golf swing technique of women shows diversity in the execution of the golf swing between the sexes (women and men) in terms of posture, grip, lower limb performance, and upper-body rotation, specifically in the range of thorax and pelvis rotation and their interaction during the golf swing [2,3,4,5]. These differences mainly arise because of their basic physical characteristics. Men are usually heavier and taller than women, and their shoulders are broader than the hips. In contrast, women’s hips are usually wider than their shoulders, and their centre of gravity is lower than that of men. In executing the golf swing, high flexibility and balance can be an advantage for women, while men have the advantage of high strength [1]. Based on the results of their study, Egret, Nicolle, Dujardin, Weber, and Chollet [6] conclude that women should use a specific swing technique.

The studies dealing with the kinematic analysis of the golf swing performance outline the following decisive parameters of upper-body movement: maximum upper torso (thorax) rotation, maximum pelvis rotation, X-factor, and X-factor stretch [7,8,9,10,11]. The X-factor is described as the angular difference between the thorax and pelvis at the top of the backswing, and numerous studies have dealt with its influence on club head velocity at impact [12,13]. The X-factor stretch is the phase when the pelvis starts its rotary motion toward the target, that is, against the movement of the thorax, which increases the X-factor, which then affects the speed of the club head at contact with the ball [8,10]. The indicator of the energy produced during the golf swing, which is the main determinant of the resulting distance of the ball’s flight, is the golf club velocity (GCV) at impact [14,15]. It is known that during the swing motion, the thorax rotates more than the pelvis [7,16,17].

Sex differences in golf swing execution were reported for adult golfers and showed higher range of motion in women than in men but higher GCV for men than women. Egret et al. [6] found the following average values of the thorax rotation at the top of the backswing: 84.1° in men and 109.4° in women. The average rotary values of the pelvis in males and females were 37.9° and 63.9°, respectively. Zheng, Barrentine, Fleisig, and Andrews [18] showed that the relative difference between the thorax and pelvis rotations (X-factor) was 60° in women and 58° in men, with generally higher values for both thorax and pelvis rotation in women than in men. They also stated that although they found greater changes in the angular position of the thorax and pelvis between the top of the backswing and the impact in women than in men, the effect on energy production was higher in men than in women.

Studies dealing with the kinematics of the golf swing provide data on the upper-body movement and the GCV, mostly when using the driver [18,19,20,21] and iron 5 or 7, which are so-called medium irons [15,22,23,24]. The reason for choosing the driver or a longer iron is the high demand for accurate execution of the golf swing to achieve accurate contact and correct trajectory of the ball flight; therefore, the differences between players of different levels can be extracted.

Recently, golf researchers have employed techniques to study the coordination between segments rather than the relation of kinematic measures to individual segments because the coordination measure provides better insights into the quality of a movement than the latter [25,26]. In golf, an important coordination between segments is the pelvis–thorax coupling, as this coupling is responsible for the stretch–shortening cycle involved in the golf swing [27]. In addition to other techniques, the continuous relative phase (CRP) has recently been used to quantify the dynamics of pelvis–thorax coupling [26,27]. Lamb and Pataky [27] showed that pelvis–thorax coupling generally varies among individuals. These authors also found differences in the club used as well as in the swing effort. Although the differences in the execution of the golf swing in the kinematics of the pelvis and thorax movement between women and men are well known, no study has explored those differences between sexes and its influence on the GCV in junior players. To the best of our knowledge, no study has investigated the potential sex-related differences with respect to pelvis–thorax coupling to date. Thus, the aim of the current study was to determine the differences in the golf swing execution in terms of the parameters of the pelvis and thorax movement and their relationship to the GCV between male and female junior golfers. We presume that boys will have a higher GCV than girls, as indicated by previous studies [28], due to more suitable anthropometric and somatic parameters and better physical fitness. However, a higher range of movement of the upper-body components is anticipated in girls due to higher flexibility. We aim to identify the significant correlation between the parameters of pelvis and thorax movement and the GCV for both sexes. The results of this study will provide an insight into the issue of sex-related differences in the execution of the golf swing and can lead to improved recommendations for golf swing execution teaching systems for junior golfers and for optimisation of the golf swing execution in both sexes. Furthermore, the results may highlight differences in golf swing execution between elite adult and junior players and point to the need to take into account the development of the individual when designing a training plan, especially in the area of golf swing technique.

## 2. Materials and Methods

### 2.1. Participants

Fourteen high-level junior golfers were tested (golf performance-handicap (hcp) = 1.8 ± 1.8; playing experience: 7.2 ± 1.8 years) to identify and compare kinematic parameters. Eight female players (age: 15.4 ± 1.0 years; height: 1.66 ± 0.06 m; weight: 58.9 ± 11.8 kg; hcp: 1.5 ± 2.2; playing experience: 6.6 ± 1.7 years; training per week: 19.0 ± 7.9 h) and six male players (age: 15.8 ± 1.7 years; height: 1.80 ± 0.07 m; weight: 72.8 ± 18.9 kg; hcp: 0.6 ± 2.0; playing experience: 8.0 ± 1.6 years; training per week: 22.8 ± 7.5 h) participated in this study. All participants were right-handed. At the time of testing, none of the players suffered from pain or injury and had not been injured for 6 months before testing. All the included players were selected by the team of the Czech Golf Federation (the National Team of the Czech Republic) in one of the junior categories. Voluntary response sampling was used as a sampling approach. All golfers from the national team had the opportunity to participate in the study. The research was approved by the university ethics committee and conducted in accordance with the ethical standards of the Helsinki Declaration and research in the field of sports sciences. Research participants and their legal guardians provided a signed informed consent before the start of the study.

### 2.2. Equipment

A 3D kinematic analysis [29] was carried out to examine a number of quantitative kinematic parameters of the golf swing. Furthermore, it can be used as a supporting method to determine the parameters affecting performance during the swing [16,30]. To determine the position of the pelvis, thorax, and golf club, the active markers with four cx1 scanning units of the CODA Motion System 3D Kinematic Analyser (Charmwood Dynamics Limited, Leicestershire, UK) were used with a sampling frequency of 200 Hz. In the current study, the movements of the pelvis and thorax were analyzed as angular changes in the horizontal plane; therefore, it was sufficient to use two markers to identify their position. The markers identifying the position of the pelvis were located on the anterior superior iliac spine and posterior superior iliac spine for each side. To determine the position of the thorax, the markers were placed on the right and left acromion. The position and velocity of the golf club were determined using two markers placed on the club shaft: one was placed just under the grip and the other 10 cm above the leading edge. The coordinate system was defined on the basis of the target line, which represents the X-axis in the three-dimensional system. The Y-axis was determined perpendicular to the X-axis in the horizontal plane, and the Z-axis was vertical.

### 2.3. Procedures

Testing was carried out at the end of the preparatory phase of the season under laboratory conditions. After an individual warm-up session, each player performed 10 strokes with their own driver in order to become familiar with the laboratory environment. After placing the markers on the body, each player performed five strokes with the driver to verify that his/her movement was not constrained by the placed markers. Afterwards, the players were instructed to approach each stroke as they would do so in a tournament and performed 10 measured and analyzed strokes. The players launched the ball from an artificial turf grass commonly used in driving ranges and were allowed to choose the size of the tee to suit their individual needs. The ball was placed on a rubber tee located in the mat. A net was located 4 m from the tee, and the direction of the shot was clearly marked by a vertical mark on a portable board hung on the wall. The target line formed by the tee and target mark was parallel to the X-axis in the coordinate system used. The players used their own drivers, new balls (Titleist, Pro V1) provided in the laboratory, and their own golf shoes. The laboratory set-up is displayed in Figure 1.

### 2.4. Data Analysis

The positions of the markers were recorded as a time series, and the values of the individual parameters were recorded at the following discrete events: setup position, top of the backswing, and contact with the ball (impact). These discrete events associated with the swing were defined using the kinematics of the club motion. The start of the swing was determined as the moment when the club reached a speed of 0.2 m/s in the direction away from the target and the top of the backswing as the moment when the club had zero speed during the transition from the backswing to the downswing. The moment of impact was defined as the moment of local peak acceleration when the marker located closer to the club head slowed down on contact with the ball. The parameters of the maximum thorax rotation (Tmax) and the maximum pelvis rotation (Pmax) were calculated as the angular change in the horizontal plane (plane x-y) between the start of the swing and the top of the backswing. The GCV at the impact was determined using the maximum velocity of the marker located closer to the club head just before the moment of impact. The X-factor (Xfac) was calculated as the angular difference between the thorax and pelvis in the horizontal plane (plane x-y) at the top of the backswing. The X-factor stretch (SXfac) was defined as the difference between the X-factor (identified at the top of the backswing) and the highest value of the X-factor, which usually follows the top of the backswing [8].

Further, rotations around the pelvis and the thorax Z-axis were used to calculate CRP to determine the coupling of the pelvis and thoracic rotation. Applying the approach of Lamb and Pataky [27], the centered Hilbert transform [31] was used to determine the CRP between the pelvis and thorax, which is a higher-order measure for their coupling. For this, the kinematic data of each participant’s stroke were aligned around the maximum pelvis rotation and time normalized to 200 samples. Data processing and CRP calculations were performed in Python 3.6.2, running on a Windows 7 operating system.

### 2.5. Statistical Analysis

Data are presented as means with the standard deviation. The normality of the data was verified using the Shapiro–Wilk test. No violation of normality of the data was found for any of the observed variables divided by gender (*p*-value ranged between 0.164 to 0.892). Analysis of variance (ANOVA) was used to assess the selected parameters. The effect size was evaluated using the “partial eta square” coefficient (ηp2), which explains the proportion of the variance of the monitored factor. We used Pearson’s product–moment correlation coefficient (r) to assess the association between the parameters. The strength of the relationships was described as detailed by Portney and Watkins [32], where 0.00–0.25 = little or no relationship, 0.26–0.50 = fair degree of relationship, 0.51–0.75 = moderate to good relationship, and 0.76–1.00 = good to excellent relationship. For all analyses, the statistical significance level was set at *p* < 0.05, to reject the null hypotheses. Statistical analysis was performed using IBM^®^ SPSS^®^ v21 (Statistical Package for Social Science, Inc., Chicago, IL, USA, 2012).

Statistical parametric mapping (SPM) [33] was used to statistically analyse the CRP time series to test the null hypothesis that there is no statistical difference between boys and girls with regard to pelvis–thorax coupling. The differences were assessed using a two-sample t-test. For each time point, a t statistic was calculated, resulting in a t(t)—trajectory that described the time-dependent differences between the pelvis–thorax coupling with respect to sex. Furthermore, a t(t)—trajectory threshold was determined to establish statistically significant differences when exceeded. The significance level for SPM analysis was set at 0.05. SPM analysis was conducted using the spm1d package [33] for Python. For the purpose of the SPM analysis which has not been implemented for unbalanced group sizes two girls were randomly excluded from the dataset.

## 3. Results

### 3.1. Sex Effect

Regarding the individual parameters (Table 1), it was found that the male players reached lower values in the parameter Pmax (38.96° ± 8.44°) than female players (46.80° ± 6.44°) and higher values in the parameters Xfac (males: 77.78° ± 15.12°; females: 68.16° ± 4.41°) and GCV (males: 27.31 m/s ± 1.99 m/s; females: 23.01 m/s ± 1.34 m/s).

A significant effect of sex was observed for the parameters of Pmax (F = 6.28, *p* < 0.05, ηp2 = 0.21), Xfac (F = 5.41, *p* < 0.05, ηp2 = 0.18), and GCV (F = 31.98, *p* < 0.01, ηp2 = 0.57). In the parameters Tmax and SXFac, no significant sex effect was found (Figure 2).

Figure 3a shows the mean CRP values for both boys and girls. The CRP values are negative for boys and girls, which means the pelvis led the thorax through the phase space. An SPM analysis was conducted between boys and girls (Figure 3b). The test revealed very similar results, excluding the different possible combinations of the two girls. Small differences at the beginning of the swing are probably errors that most likely occurred because of the Gibbs phenomenon [34] and were ignored. However, there is a significant difference during the backswing in the pelvis–thorax coupling between boys and girls.

### 3.2. Relationship of Pelvis and Thorax Movement with Golf Club Velocity

In female players, no significant relationship was found between thorax and pelvic movement parameters and GCV. In male players, we found a significant negative relationship between Tmax and GCV (r = −0.941, *p* < 0.01) and between the parameters Xfac and GCV (r = −0.847, *p* < 0.05) (Table 2).

## 4. Discussion

### 4.1. Sex Effect

The current study investigated sex differences in pelvis and thorax motion during the golf swing performed by elite junior players. Furthermore, a relational analysis of the individual body movement parameters and GCV was conducted. Previous studies have described the difference in the movement of individual segments of the upper-body between men and women or between players of different performance levels [11,18,21,35]. The rotation of the pelvis and thorax, X-factor, and X-factor stretch have been identified as the key parameters of the movement of the body leading to the production of maximum energy, that is, the maximum GCV [15]. Similarly, different ranges of movement have been identified in the stated parameters in men and women [21]. Tremblay et al. [36] reported significantly greater flexibility in girls than in boys in three age categories (8–10, 11–14, and 15–19 years). We predicted higher ranges of movement in the abovementioned parameters in girls than in boys; however, we predicted a higher GCV in the boys because boys of the junior age group (14–19 years) have higher muscular strength than girls [36,37]. One of the main determinants affecting the difference in the level of muscular strength between the sexes is body weight and the amount of muscle mass [37]. A previous study on the kinematics of the swing of professional golfers indicated a higher GCV at the moment of impact in males than in females [18]. Based on previous findings, we predicted a positive or no relationship between the different parameters of body movement and GCV in both sexes.

The girls in the present study reached a significantly higher range of motion in the pelvis rotation parameter compared to the boys; however, we did not find significant differences in thorax rotation. The boys achieved significantly higher values for the X-factor and GCV. A higher range of pelvic rotation in females than in males has also been found in previous studies [18]. Zheng et al. [18] stated that the range of pelvic movement for male professional players was 42.0° ± 7.0° and for female professional players was 49.0° ± 8.0°, which is approximately 3° higher for both men and women than the participants in our study (boys: 39.0° ± 8.4°; girls: 46.8° ± 6.4°). The small difference might be due to the fact that in our study 2D angles in transverse plane were used while Zheng et al. [18] used 3D angles. Myers et al. [10] reported pelvic rotation at the top of the backswing of adult golfers in three groups based on ball velocity (high: 44.9° ± 10.3°, middle: 47.5° ± 17.4°, and low: −49.9° ± 11.4°), which was higher in all groups than the boys in our study and also higher in the middle and low ball velocity groups than girls in our study.

Regarding pelvis–thorax coupling, we found a significant difference during the backswing between boys and girls. In boys, the backswing phase begins with thorax rotation followed by the pelvis, but in girls, the backswing starts with simultaneously rotation of thorax and pelvis. However, in the downswing phase, the more important part of the swing, when the energy associated with the GCV is produced, there was no statistical difference in pelvis–thorax coupling between the boys and girls. Considering the above result, the high GCV of the boys can be attributed to their physiological composition and its changes during maturation such as the rapid development of muscle strength [36], rather than pelvis–thorax coupling. This is a very interesting result which, to the best of our knowledge, has not been investigated before.

In summary, our results show that girls use their pelvis when swinging and secure rotational movement using their lower body to a greater extent than boys. This is in accordance with the technique recommendations for girls and women found in the current teaching literature [38].

### 4.2. Relation of Upper Body Movement and Golf Club Velocity

The analysis of the relationship between the club head velocity and the pelvic and thorax movement parameters did not show any significant relationship in the junior girls category. In boys, we found a significantly negative relationship between the GCV and the parameters of maximum thorax rotation (r = −0.941, *p* < 0.01), and maximum X-factor (r = −0.847, *p* < 0.05). In previous studies on the relationship between the GCV and the parameters of body movement in adult individuals, there was no negative relationship between Tmax and GCV and the X-factor and GCV. This trend may be due to the age of the tested players, for example, in boys aged 14–19, there is a rapid development of strength capabilities [36]. In skilled adult golfers and professionals, it is known that the higher the X-factor, the higher the GCV at impact [8,10,39]. For example, Myers et al. [10] reported a significantly higher X-factor at top of the backswing in players with high ball velocity (59.1° ± 8.2°) compared to those with low ball velocity (44.2° ± 7.7°) and suggest, that the X-factor at the top of backswing is ultimately contributing to increased ball velocity, which is related to GCV. Interestingly, in the current study involving elite junior golfers, we found a contrary relationship between X-factor and GCV, particularly in boys. This negative relationship may be influenced by the biological development of the individuals, which gradually leads to anthropometric and morphological differences resulting in increased body weight, increased muscle strength, production of androgen hormones, and other changes. An important factor in increases in muscular strength during adolescence is the effect of endocrine adaptation during maturation, such as increased levels of androgens and growth hormones [37]. Ramos et al. [37] reported increasing levels of testosterone and growth hormone during childhood in boys (12, 14, and 17 years old) compared to girls. At the age of 14, we can talk about the beginning of the elite golf training, and at about the age of 16, we can begin to discuss the performance level [40] reflecting a consistent golf swing technique. The increase in performance in both sexes in the junior age is closely related to the achieved length of the individual shots, especially while playing with the driver [41,42] and its correspondence to a high GCV at impact. These changes in GCV are achieved by the improvement in the fitness parameters (increase in muscle strength, muscle firming up, and reduction of flexibility in men) [42], which are achieved not only by hormonal changes but also by targeted fitness training. These unique findings should be further explored in future studies, where we would recommend focusing on longitudinal monitoring of kinematic parameters in the elite junior golf players category (14–20 years) with the determination of biological age, as differences of physical development might be up to 4 years compared to chronological age [43] and might therefore affect golf swing performance.

However, we are aware of the limitations of this study, which may include the small number of subjects in each group (girls and boys) and that although the research sample is homogeneous in terms of performance, only elite players of junior age were recruited and those non-elite were not presented. Additionally, only 2D angles in transverse plane were established in this dataset and thus the representation of the position of the thorax and pelvic in 3D space is missing, e.g., shoulders tilt which is closely related to the forward bend and pelvis tilt related to lower limb function during the golf swing. Furthermore, the kinematic analysis method used with the placement of the active points directly on the participant’s skin may have caused slight discomfort and, as a consequence, a golf swing execution that may not match the actual execution used during play or practice. In the future research aiming to understand thorax and pelvic movements during the golf swing in junior golfers, a more holistic approach might be applied, including, for example, the methods for determining biological age and muscle strength.

## 5. Conclusions

In the current study, the results show sex-related differences in golf swing execution in pelvic and thorax movement parameters in junior golfers. Whilst being aware of the study limitations, we suggest that, when teaching the golf swing to juniors, it is necessary to respect both the differences between the sexes (higher range of pelvic rotation in girls but higher X-factor in boys) and the level of physical preparedness between individuals, especially the range of motion of the pelvic and thorax during the golf swing. In addition, different strategies appear during the beginning of the backswing between sexes, where boys control the initial movement with the thorax and girls with the thorax and pelvic simultaneously. Further, we found a negative correlation between X-factor and GCV in boys of junior age. Based on these unique findings, we suggest that coaches should use a tailored approach when teaching the golf swing technique not only between sexes, but also between individuals of the same sex.

## Figures and Tables

**Figure 1 sports-11-00060-f001:**
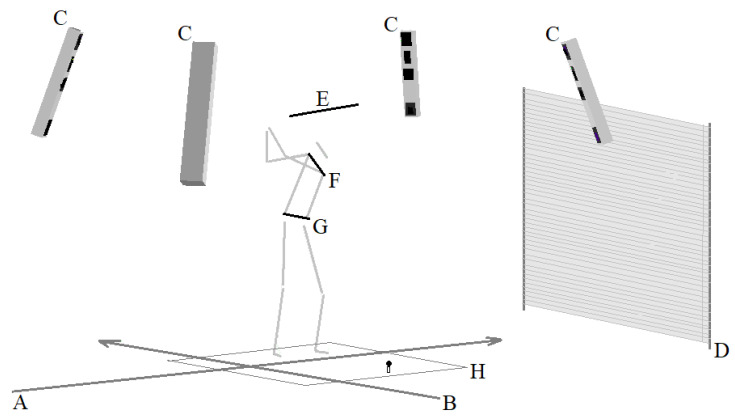
Display of the laboratory set-up—viewed at the top of the backswing. Legend: A, X-axis; B, Y-axis; C, CODA Motion cx1 scanning units; D, golf net; E, golf club; F, thorax position; G, pelvis position; H, an artificial turf grass with the ball on the tee.

**Figure 2 sports-11-00060-f002:**
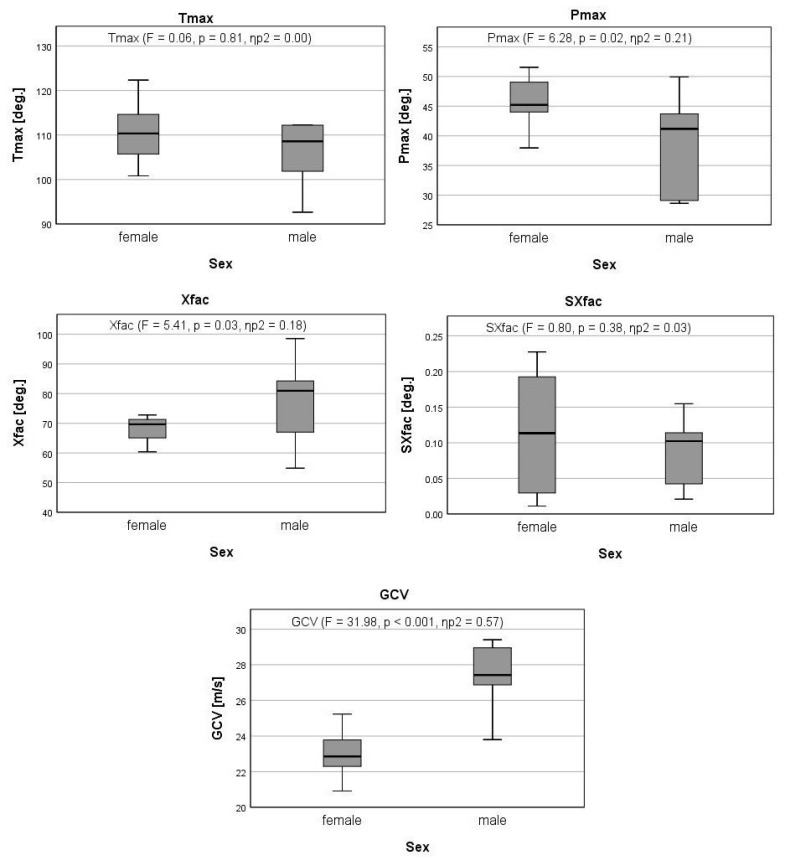
Sex effect in kinematic parameters. Legend: Tmax, maximal thorax rotation; Pmax, maximal pelvic rotation; Xfac, X-factor; SXfac, X-factor stretch; GCV, golf club velocity at impact. Boxplots show interquartile range (IQR) between Q1 and Q3. Black lines in the box plots indicate median. The whiskers outside the boxplots indicate minimum and maximum values.

**Figure 3 sports-11-00060-f003:**
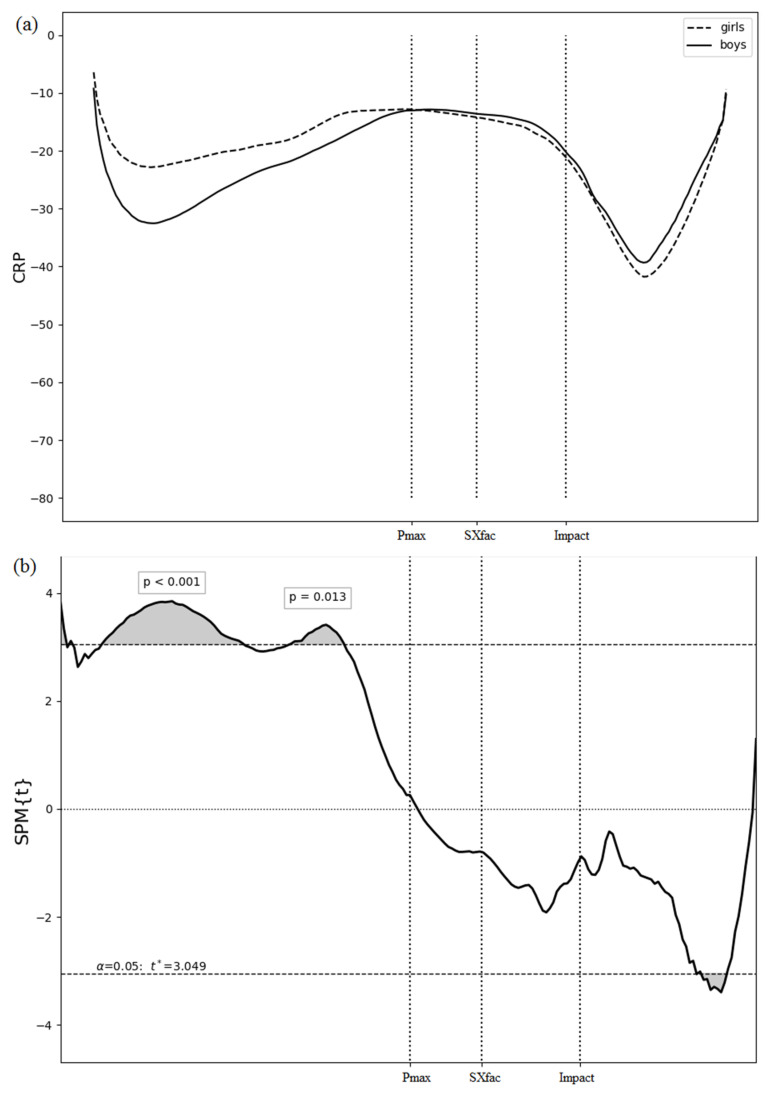
(**a**) The mean CRP values of the boys (solid line) and girls (dashed line); (**b**) t(t) trajectory (black) and corresponding critical thresholds (horizontal dashed black); vertical dashed lines represent the start of the transition phase when the pelvis change direction, the mean moment when the angle between pelvis and thorax is maximal, and the impact.

**Table 1 sports-11-00060-t001:** Descriptive statistics of body movement parameters and golf club velocity.

Parameters		Sex	Mean	Standard Deviation
Body movement parameters	Tmax	Female Male	110.57°	7.21°
			111.34°	17.51°
	Pmax	Female Male	46.80°	6.44°
			38.96°	8.44°
	Xfac	Female Male	68.16°	4.41°
			77.78°	15.12°
	SXfac	Female Male	0.11°	0.09°
			0.09°	0.05°
Golf club	GCV	Female Male	23.01 m/s	1.34 m/s
			27.31 m/s	1.99 m/s

Legend: Tmax, maximal thorax rotation; Pmax, maximal pelvic rotation; Xfac, X-factor; SXfac, X-factor stretch; GCV, golf club velocity at impact.

**Table 2 sports-11-00060-t002:** Relational analysis between upper body movement parameters and golf club velocity.

Correlations					
		Tmax	Pmax	Xfac	SXfac
GCV Girls	Pearson Correlation	0.375	0.277	−0.070	−0.168
	Sig. (2-tailed)	0.359	0.507	0.870	0.691
GCV Boys	Pearson Correlation	−0.941 **	−0.202	−0.847 *	−0.489
	Sig. (2-tailed)	0.005	0.700	0.033	0.329

Legend: Tmax, maximum thorax rotation; Pmax, maximum pelvic rotation; Xfac, X-factor; SXfac, X-factor stretch; GCV, golf club velocity at impact; * *p* < 0.05; ** *p* = < 0.01.

## Data Availability

Publicly available datasets were analyzed in this study. This data can be found here: https://zenodo.org/record/7233465#.Y1JRAXZBybh, accessed on 21 October 2022.

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
