# Peer review of "Do the Pelvic and Thorax Movements Differ between the Sexes and Influence Golf Club Velocity in Junior Golfers?"

_sports, 2023, doi:10.3390/sports11030060_

Round 1

Reviewer 1 Report

This study investigated the association between gender and torso movements as well as golf club velocity in junior golfers. While the study is interesting, the English is difficult to read and the statistical analysis is also flawed. The overall study design and goal is very confusing. Extensive revisions are necessary before the manuscript is ready for publication. Here are some specific comments.

The title is clumsy, recommend retitling as “Do pelvis and upper torso movement associate with sex and influence golf club velocity in junior golfers?” Only any title which sounds simpler.

The overall goal and study design are confusing. The title was on sex, pelvis and upper torso movements, and golf club velocity; the abstract was on variational analysis of golf, upper body movements; (upper body is different from upper torso); the objective section (i.e. last paragraph of the introduction) mentioned component factors of upper-body movement; variational analysis; coupling; in the Methods, the analysis used MANOVA, correlation and CRP. All these were scattered and kept readers from understanding how the study was designed.

For the statistical analysis, the authors mentioned that they used Multilevel Analysis of variance (MANOVA). In fact, MANOVA is a multivariate analysis of variance and multilevel analysis is another type of statistics. Unfortunately, the results in the paper neither demonstrated any results on Multilevel analysis of variance nor MANOVA.

Line 13: 10 trials of what

Line 14: What are the parameters

Line 15: upper body coordination is not a parameter allowed for statistical parametric mapping

Line 16 – 22: P-value shall be presented in full unless p < 0.00.1

Line 22: The study is about the biomechanics of sports, while the implications mentioned hormones and biological development. Please link them up. It does not sound related.

Line 28 – 31: English not understood, and I suppose reporting existing literature shall use past tense. English revision throughout the manuscript is necessary.

Line 36 – 41: The paragraphs mentioned that males and females were different and therefore women should use a specific swing technique. If gender plays a role, both sexes shall use gender-specific technique. Why only women?

Line 45: Please move the definition of X-factor after this sentence first, since some readers may not understand X-factor at the first place.

Line 55: Please start with the a topic summary of this paragraph.

Line 64: Please make sure the word “upper body” and “upper torso” has been described correctly in text. For the findings or goals or setting of this manuscript that used the term “upper body/torso”. Please be consistent with the terminology.

Line 77-79: Please describe why is it useful or necessary to formulate this study before addressing the “no-body do that”.

Line 83 – 85: Could not be understood, especially “component factor”, “variation”, which contradicted with the overall aim of this study.

Line 86 – 87: Please justify the premise for the presumption

Line 90 – 93: Please elaborate more on the pathway to impact

Line 93 – 103: This context belongs to the template file. Please remove them

Line 106: What is hcp?

Line 106: Please justify the sample size estimation

Line 106: Please include more information on the recruitment process and sampling approach (is it randomized).

Line 115: Please repeated the ethical considerations here (e.g., informed consent, ethical approval with reference number)

Line 116: Shall be “equipment”

Line 125: Basically, at least three markers are required to represent a body segment. Only two markers were presented here for the upper torso. Please justify.

Line 128: What is cx1?

Line 129 – 131: These words are not informative. Y axis is always perpendicular to X axis. Consider revise x and y into anteroposterior, or mediolateral axes in the result section

Line 151: Please superscript the velocity unit

Line 146: The participants performed 10 trials. The authors did not mention how to take care of the data reduction process of the 10 trials.

Line 172: Data were presented as mean and standard deviation (is sufficient)

Line 175: The method mentioned here was “eta square” but the results mentioned “partial eta square”. Which is correct?

Line 172: For all the statistical analysis here, including the MANOVA(or multilevel, not sure), Pearson, t-test, and parametric mapping. There is a basic requirement, including data normality and no outliers. Please include the checking process and justification.  

Line 190: Shall be the “significance level” or “alpha level”

Line 193: Descriptive statistics shall be presented first in the results.

Line 193: I cannot see any results of MANOVA here.

Table 1: It deems inappropriate to just paste the statistical table from the software into the results. Please consider using a boxplot of two sexes for the five parameters that illustrate the distribution and significance level in the plot.

Line 212: The implementation issues of SPM shall be put in the Method section. Instead of mentioning that two girls were excluded (which created a lot of confusion). Please consider mentioning that 6 females were randomly sampled from the dataset to achieve a balanced dataset for the SPM.

Table 3: Similarly, it deems inappropriate to just paste the statistical table. Please consider using a scattered plot with regression line and showed the Coefficient and p-value inside the plot.

Line 276: Please expand the logic here. I cannot find the relationship between club velocity and physiology. Honestly, the discussion on this issue is very important and please present the logic clearer.

Line 286: If this is the case, the authors shall measure the muscular strength of the participants. Please make discussion on it. Frankly, this may not be the case among athletes. The authors may need supporting references for this.

Line 308: Do the participants confined to professional athletes or amateur athletes? The external validity with regard to their class or training time shall be addressed.

Line 309 – 311: Not understood

Line 317: I did not find any statistical analysis relating to the range of pelvic movement and upper torso

Line 321 – 327: These are not conclusions. Please discard them and retain them in the discussion.

Author Response

Dear reviewer, point-by-point response is uploaded as a Word file. Thank you for your comments and recommendations

Reviewer 2 Report

Please provide an abstract with more structure: introduction, methods with participants, intervention and outcome .. results and conclusions

What kind of participants did you include (not the number, it's a result). Did you use markers and optoelectronic cameras? Evaluation scales?

Club speed should be described in introduction or/and outcome methods, it may not be so easy to interpret for all readers of the journal ... let them understand the importance of study

Male / female

What is the X-factor

Lines 28-29 contradict 32-34, write that at an amateur level what you said before is not so true .. I don't know

41 there is no solid justification in the literature for these purely anatomical causes

45 what is the x-factor

53 also describe the club head speed in detail here

88-93 only put goals, not what you have examined

106 Methods: Hcp? However these are all results, in the methods it must be described what kind of population you wanted to reach.

117 might you put a figure?

133 and 146 Are there references in the literature?

193 Enter the characteristics of the participants

Gender instead of Sex ..

Tbale 3 lacks the meaning of the asterisks

Soften the conclusions, the sample is small .. the differences are of gender instead of sex .. I would emphasize that they are adolescents at most young adults. It is a concept not to be underestimated

Author Response

(The authors gave the same response as above.)

Round 2

Reviewer 1 Report

Thanks so much for the authors for their hard work on modifying and greatly improving the manuscript. I have two following up issues on the manuscript. 

- I only see comparison between two groups (male vs. female). I doubt whether the statistical analysis shall be independent sample t-test, instead of ANOVA. 

- Line 112, the authors mentioned that their study can highlight the differences between elite adult and junior players. I did not see any comparison of elite adult and junior players in the analysis. The objective (statements) shall be confined to the findings of the results to avoid confusion, while additional exploration could be included in the discussion. 

- For the external validity in discussion, external validity is always the limitations that cannot be maintained since every experiments just extracted a small sample from the population and could not "fully" represent the population. It is better to discuss the limitations in which the part that "the sample could not well-represent" the population.  For example, the study only recruited professional athletes and those non-professionals were not presented. 

Author Response

Dear reviewer, point-by-point response is uploaded as a Word file. Thank you for further comments and recommendations. Native speaker carefully read and stylistically edited those parts where necessary.

Reviewer 2 Report

Dear Authors, I must congratulate you on the effort you put into the review. I just want to leave a minor suggestion: since the procedure is not present in the literature, can you provide a figure, a photo? For the reproducibility of the manuscript and the methodology used..

Author Response

Dear reviewer, thank you for further suggestion. Figure was added into the manuscript and the response is uploaded as a Word file. 
